# Contribution of Yeast Studies to the Understanding of BCL-2 Family Intracellular Trafficking

**DOI:** 10.3390/ijms22084086

**Published:** 2021-04-15

**Authors:** Akandé Rouchidane Eyitayo, Mathilde Gonin, Hubert Arokium, Stéphen Manon

**Affiliations:** Institut de Biochimie et de Génétique Cellulaires, Université de Bordeaux, CNRS, UMR 5095, 1 Rue Camille Saint-Saëns, 33077 Bordeaux, France; akande.rouchidane-eyitayo@ibgc.cnrs.fr (A.R.E.); mathilde.gonin@ibgc.cnrs.fr (M.G.); roaks14@yahoo.fr (H.A.)

**Keywords:** mitochondria, Mitochondria-Associated Membranes, Bcl-2 family, endoplasmic reticulum, apoptosis, Translocase of Outer Membrane

## Abstract

BCL-2 family members are major regulators of apoptotic cell death in mammals. They form an intricate regulatory network that ultimately regulates the release of apoptogenic factors from mitochondria to the cytosol. The ectopic expression of mammalian BCL-2 family members in the yeast *Saccharomyces cerevisiae,* which lacks BCL-2 homologs, has been long established as a useful addition to the available models to study their function and regulation. In yeast, individual proteins can be studied independently from the whole interaction network, thus providing insight into the molecular mechanisms underlying their function in a living context. Furthermore, one can take advantage of the powerful tools available in yeast to probe intracellular trafficking processes such as mitochondrial sorting and interactions/exchanges between mitochondria and other compartments, such as the endoplasmic reticulum that are largely conserved between yeast and mammals. Yeast molecular genetics thus allows the investigation of the role of these processes on the dynamic equilibrium of BCL-2 family members between mitochondria and extramitochondrial compartments. Here we propose a model of dynamic regulation of BCL-2 family member localization, based on available evidence from ectopic expression in yeast.

## 1. Introduction

BCL-2 family members are major regulators of the apoptotic process in mammalian cells [1,2,3] (for reviews). These proteins are largely conserved in vertebrates [4] but closely related homologs have been identified in most invertebrates [5,6,7]. However, no BCL-2 homolog has been identified in plants, where the ectopic expression of murine pro-apoptotic protein tBID does not increase Programmed Cell Death [8]. Although the yeast *Saccharomyces cerevisiae* does not contain any obvious BCL-2 homolog (but see the discussion on Ybh3/Bxi1 below), the expression of mammalian BCL-2 family members recapitulates events that are associated with apoptosis (reviewed in [9]), making yeast an adequate tool to investigate molecular aspects of the functions of these proteins.

Their main function is to regulate the permeability of the outer mitochondrial membrane (OMM) towards so-called apoptogenic factors that are normally sequestered in the mitochondrial intermembrane space (IMS) [10,11] (for reviews). The pro-apoptotic proteins BAX and BAK form the core of the permeabilization process: following an apoptotic signal, they oligomerize and form and/or stabilize a large size pore that enables the release of apoptogenic factors to extramitochondrial compartments [12] (for review). BAX/BAK function in OMM permeabilization is prevented by anti-apoptotic proteins (BCL-2, BCL-xL, MCL-1...) and directly or indirectly activated by BH3-only proteins (tBID, BIM, PUMA, NOXA, BAD...).

The subcellular localization of BCL-2 family members is a highly dynamic and highly regulated process. Typically, BAX was initially found in the cytosol of non-apoptotic cells and was translocated to the OMM following the induction of apoptosis. This rather simple view was challenged by the demonstration that BAX could be retrotranslocated from mitochondria to the cytosol [13]. Interestingly, anti-apoptotic proteins BCL-2 and BCL-xL have been shown to stimulate both BAX translocation [14,15] and retrotranslocation [13,16,17] and the subcellular localization of these two proteins is also multiple and regulated [18] (for review). Besides their well-established function as regulators of OMM permeability, they contribute to the regulation of Ca^2+^ regulation in the ER (Endoplasmic Reticulum) and the modulation of the activity of some transcription factors in the nucleus. These different localizations (and function) might be related to post-translational modifications of these proteins, such as phosphorylation for BCL-2 [19] and phosphorylation [20] and deamidation [21] for BCL-xL.

It is now well established that mitochondria and ER come close to each other in regions that have been termed MERC (Mitochondria-ER Contacts) [22]. Recently, an interactomic study showed that BAX was present in these domains, characterized as proteins that are near a linker between mitochondrial protein TOM20 and ER protein SEC61 [23]. Recent studies also supported the hypothesis that ER domains participating to these contacts, which are termed Mitochondria-Associated Membranes (MAM), are involved in the regulation of the localization of BAX [24] and BCL-2 [25]. Interestingly, components of the outer mitochondrial membrane sorting complex, TOM20 and TOM22, have been found to be involved in BCL-2 [25,26] and BAX [27,28,29] mitochondria translocation, respectively. However, it should be noted that the alteration of BAX/TOM22 interaction did not impair BAX mitochondrial localization, but rather limited its capacity to promote an efficient permeabilization of the OMM. Another study showed an interaction between the BH3-only protein BIM, one of the main activators of BAX, with TOM (Translocase of Outer Membrane) components but the functional role of this interaction remains unclear [30]. Besides TOM components, an interaction between BAK and SAM (Sorting and Assembly Machinery) components metaxins 1 and 2 was demonstrated, with a possible involvement in the activation of BAK, in association with VDAC2 [31,32]. Other studies suggested that MCL-1 [33,34] interacts with OMM proteins. Although they are not of proteinaceous nature, the well-established interaction between tBID and cardiolipin might also be included in these studies [35,36]. The link between the localization of BCL-2 family members and the dynamics of the interaction between the OMM and other membrane compartments thus requires further investigations.

The yeast *Saccharomyces cerevisiae* is a long-used model for the study of conserved eukaryotic cells functions. Due to its nature of facultative aerobe, it is particularly interesting in studies related to mitochondria. Indeed, opposite to mammalian cells, yeast can survive to dramatic alterations of mitochondrial functions. It is, therefore, possible to investigate the consequences of major mitochondria alterations on other cellular processes. This unique *S. cerevisiae* characteristics led to the design of tools aiming at using it as a model for studying the action of BCL-2 family members on mitochondria [37,38,39] (for reviews). Although yeast displays several forms of programmed cell death (PCD), these processes do not involve any BCL-2-like proteins, which are absent from the yeast genome. A protein containing a BH3 domain, Ybh3, has been identified [40] and is the same protein as the BAX-Inhibitor protein (Bxi1) that had been previously identified as an ortholog of BAX inhibitors of plants [41]. The deletion of Ybh3/Bxi1 decreased yeast resistance to heat shock, through decreasing the Unfolded Protein Response (UPR) and Ca^2+^ signaling [41]. It also decreased yeast PCD induced by oxidative stress and other stimuli, including the ectopic expression of BAX [40]. Conversely, Ybh3 overexpression induced a yeast PCD that was reportedly sensitive to the ectopic co-expression of BCL-xL [40]. It is likely that at high concentration, Ybh3 and BCL-xL can interact, although it has not been directly demonstrated.

Many experimental conditions have allowed investigators to study different forms of Programmed Cell Death in yeast, which have common features, but also significant differences, with mammalian apoptosis (reviewed in [9]). The heterologous expression of mammalian BCL-2 family members in yeast was initially designed as a simple tool to investigate interactions between these proteins by the two-hybrid method [42,43]. However, it was quickly observed that the expression of the fusion proteins had functional consequences on yeast growth and viability: indeed, the expression of a fusion protein between BAX and the DNA binding domain LexA inhibited yeast growth and this effect was prevented by the co-expression of a fusion protein between BCL-2 and the trans-activation domain B42 [42]. It was further observed that the same effects on yeast growth could be observed with genuine BAX, BCL-2, and BCL-xL, which could provide information on how these proteins interfere with mitochondria [44,45,46].

It has also been observed that the ectopic expression of BCL-2 protected yeast cells against oxidative stress [47]. Interestingly, the ectopic expression of BCL-xL protected yeast against cell death but did not decrease the oxidative stress [48], thus suggesting a different action of these anti-apoptotic proteins in yeast. All these observations lead to the development of research studies aiming at using yeast as a model to gain knowledge of mammalian BCL-2, including the identification of new partners of BCL-2 family members [49,50,51] (for reviews) and more direct studies aiming to understand how BCL-2 family members cooperate to modulate OMM permeability and other processes involved in cell death [37,38,39,52,53] (for reviews).

In the present review, we will focus more specifically on yeast studies that have aimed at studying processes and proteins that are involved in the subcellular localization of BCL-2 family members, namely pro-apoptotic BAX and anti-apoptotic BCL-xL, and that contribute to the regulation of the dynamics of this localization, which is closely dependent on the dynamics of intracellular membranes.

## 2. Intrinsic Molecular Determinants of BCL-2 Family Members Localization

The usual subdivision of mammalian BCL-2 family members is based on a combination of their function (pro- or anti-apoptotic) and of their primary structure, i.e., the presence of BCL-2-Homology domains (BH1 to BH4) [54] (for a classic review). This classification distinguishes 3 sub-families:-Anti-apoptotic proteins, which includes BCL-2, BCL-xL, MCL-1, et al.-Multidomain pro-apoptotic proteins, which includes BAX, BAK, and BOK.-BH3-only proteins, which includes BID, BIM, BAD, PUMA, NOXA, et al.

Structural data have demonstrated, without any ambiguity, that anti-apoptotic proteins, multidomain pro-apoptotic proteins, and BH3-only protein BID are structural homologs [55], which thus could be termed as “BCL-2 homologs” (to distinguish them from “BCL-2 family”) [5]. Other BH3-only proteins do not share these structural features. BIM, BAD and BMF are intrinsically disordered proteins that become only partly structured when their BH3 domain is bound to a BCL-2 family partner [56]. This is even more spectacular for PUMA, of which the BH3 domain can bind to its partner MCL-1 even when proline residues are introduced to break its α-helical structure [57]. On this basis, these BH3-only proteins are not structurally related to BCL-2. Proteins carrying a BH3 domain are not always directly involved in the regulation of apoptosis. A typical example is the protein BECLIN-1, which was first identified as a protein interacting with BCL-2 (hence its name) [58,59] but is a major regulator of autophagy, homolog to the yeast protein Atg6 [60]. It is noteworthy that a significant number of proteins with a BH3 domain are regulators of autophagy. Besides the widely characterized BECLIN-1, the protein NIX/BNIP3L has been characterized as a regulator of mitophagy [61,62]. More recently, the previously characterized BCL2L13/RAMBO [63] has been shown to be a functional homolog of yeast mitophagy receptor Atg32 [64].

The analysis got more complicated with the identification of more distantly related proteins with a BH3 domain. A computational analysis of these proteins showed that the BH3 domain could be present in structurally and phylogenetically distant proteins [5]. Both the weaker conservation of BH3 residues and the positioning of the BH3 domain within the whole sequence of these proteins led to the proposal that their BH3 domain is “non-canonical” and could result from a convergent evolution rather than from the divergence from a common ancestor (unlike BCL-2 homologs) [5].

This does not imply that non-canonical BH3-only proteins do not contribute to the regulation of BCL-2 homologs. Indeed, following overexpression, these domains could interact with BCL2 homologs, albeit with a lower affinity than canonical BH3 domains. As discussed above, the yeast Bxi1/Ybh3 protein, which has a non-canonical BH3 domain [5], can be inhibited by BCL-xL [19].

The functions of BCL-2 homologs involve their interaction with membranes and, more specifically, the outer mitochondrial membrane (OMM). Other proteins with non-canonical BH3 domains, such as Nix and BCL2L13, are also inserted in OMM, although others such as BECLIN-1 and IP3R are in the endosomal network). Interestingly, both BCl-2 and BCL-xL, also display extramitochondrial localization in the ER and the nuclear membrane, with specific, non-apoptotic functions associated [18] (for a review). It is then crucial to understand how these proteins interact with these membranes.

### 2.1. Role of the Hydrophobic C-Terminal α-Helix BCL-2, BCL-xL, and BAX

Except for BID, BCL-2-like proteins, whether anti- or pro-apoptotic, have a C-terminal α-helix that is largely hydrophobic that was tentatively assimilated to a membrane anchor [65]. It was therefore tempting to assume that this helix was directly responsible for the localization of these proteins to membranes. This has been confirmed, in cellulo, for BCL-xL, for which the C-terminal helix is necessary and sufficient for OMM localization [66]. The C-terminal of BCL-2 is required for membrane localization, but the specific OMM localization also depends on other factors, one of which has been identified as its interaction with the mitochondrial receptor TOM20 [25,26]. Mutants of both BCL-2 and BCL-xL deprived of the C-terminal helix lost their ability to be co-fractionated with any membrane compartments of mammalian cells [66]. However, they might still be recruited to membranes through the interaction with OMM-localized partners (such as BAX) but, in this case, they might not be inserted and may be removed by a chaotropic treatment in the absence of a detergent [66].

When expressed alone in yeast, as in mammalian cells, the anti-apoptotic protein BCL-xL is largely associated with a crude mitochondrial fraction and the deletion of the C-terminal α-helix (BCL-xLΔC) impaired this association [67]. Interestingly, a single substitution A221R, introducing a positively charged residue within the helix, was sufficient to impair the mitochondrial localization of BCL-xL, both in yeast [68] and in mammalian cells [69]. This single-substituted mutant displayed a strong alteration of its localization, in the sense that it was fully excluded from mitochondria, opposite to BCL-xLΔC [15]. This was likely due to the fact that its effect on the three-dimensional structure of BCL-xL was less drastic and less subject to side effects, such as the exposure of hydrophobic residues in the protein core that should not normally be exposed.

The localization of BCL-2 in mammalian cells is more shared between different compartments, in accordance with its distinct functions as an inhibitor of BAX (at the mitochondrial level) and as a regulator of Ca^2+^ fluxes (at the ER level) [18] (for review). It is noteworthy that a splicing variant of BCL-2, termed BCL-2β, lacks the C-terminal α-helix with a short 9 residues sequence instead; however, the localization and function of this isoform remain unclear [70] (for review). When expressed in yeast, BCL-2 also displays multiple localizations, in mitochondria and post-mitochondrial fractions. Interestingly, as in mammalian cells, the mitochondrial localization of human BCL-2 in yeast is increased by the overexpression of yeast or ectopically expressed human TOM20 [25]. Furthermore, a peptide derived from the binding sequence of human TOM20 with BCL-2 can modulate the mitochondrial localization of BCL-2 in yeast [25].

Opposite to BCL-2 and BCL-xL, the localization of BAX in mammalian cells is not predominantly mitochondrial in non-apoptotic cells. After an apoptotic signal is perceived by the cell, the localization of BAX dramatically changes, with an increased mitochondrial localization [71,72]. However, as we will see below, BAX movements between extramitochondrial and mitochondrial compartments are not unidirectional, and a dynamic equilibrium of BAX localization between different compartments is maintained. C-terminal anchored OMM-localized proteins have a hydrophobic C-terminal α-helix flanked by positive charges both at their N-side and C-side [73]. BCL-2 family members present this type of motive. However, there are several differences between the pro-apoptotic protein BAX and the anti-apoptotic proteins BCL-xL and BCL-2, in the repartition of charges flanking the hydrophobic segment. The comparison of the behavior of peptides encompassing the C-terminal domains of BCL-2 and BAX showed that BCL-2 C-ter was able to adopt spontaneously a α-helical structure in the presence of PC bilayers (provided they have an appropriate thickness) while BAX C-ter required the presence of negatively charged lipids (PG or PI) to adopt such a structure [74]. The explanation for these differences might rely on the charges: BAX displays a KK doublet on the C-side of the hydrophobic domain and no charge on the N-side. BCL-2 displays a HK sequence on the C-side and a K on the N-side. BCL-xL displays a RK sequence on the C-side and a highly charged sequence ESRKGQERFNR on the N-side. The presence of positively charged residues on the cytosolic side of the hydrophobic helix of BCL-2 and BCL-xL might explain the difference of behavior with BAX, since these charges might stabilize the membrane embedment of this helix by interacting with negatively charged lipid heads.

As a matter of fact, the C-terminal domains of BAX and BCL-xL are not interchangeable: while the C-terminal domain of BCL-xL is able to drive the mitochondrial localization of BAX in vitro, the C-terminal domain of BAX is not able to drive the mitochondrial localization of BCL-xL, both in mammalian and yeast mitochondria [75,76].

Another remarkable feature of the C-terminal domain of BAX is the presence of a Serine residue at the position 184 that is the target of several kinases (see below). The deletion of this residue (ΔS184) leads to a protein having a constitutive mitochondrial localization, indicating that this residue may play a crucial role in the localization of the protein [77,78]. Similarly, substitutions S184A or S184V lead to an increased mitochondrial localization [79], while substitutions S184D or S184E led to a decreased mitochondrial localization [80]. These different behaviors also occurred when these mutants are ectopically expressed in yeast [81]. However, the consequences on BAX activity (i.e., the capacity to release cytochrome c) were not quite as expected. Indeed, the actual activities of these BAX mutants were also dependent on their cellular content that depends on their stability [82]. Although BAX-S184V was found to be stable over a time scale of 4 h, BAX-S184D was rapidly degraded. This degradation was largely prevented by the general protease inhibitor PMSF (Phenyl methylsulfonyl fluoride) and by the deletion of Pep4, the yeast ortholog of mammalian Cathepsins, suggesting that this degradation essentially occurs in the vacuolar/lysosomal compartment. Consequently, the cellular content of BAX-S184D was much lower than the cellular content of BAX-S184V. The overall activity of these two mutants, measured at the cellular level, was in fact close to each other, but this was associated with a much higher mitochondrial content of BAX-S184V compared to the mitochondrial content of BAX-S184D. This suggested that the specific activity of BAX-S184D was much higher. This is an important observation showing that the overall capacity of BAX to permeabilize the OMM not only depends on its intrinsic ability to form pores, but also on its cellular content/stability. This is in line with a more recent study made in MEF [83].

Anti-apoptotic proteins, such as BCL-xL, add another level of regulation of the activity of these two BAX mutants [68]. In yeast, the co-expression of BCL-xL had about the same effect on the mitochondrial localization of BAX-S184D and BAXS184A (which behave like BAX-S184V). However, the co-expression of a truncated mutant of BCL-xL, BCL-xLΔC, which stimulates BAX translocation [15] but not BAX retrotranslocation [15,16], did increase the mitochondrial localization of BAX-S184D (like that of BAX-WT), but not the mitochondrial localization of BAX-S184A. Indeed, BCL-xL was more prone to retain BAX-S184D out from the mitochondrial membrane than to retain BAX-S184A [38]. Taking all the evidence into account, despite having a greater intrinsic activity when measured alone in yeast, the mutant BAX-S184D might be less active than BAX-S184A when expressed in mammalian cells because of (1) a lower stability/content, (2) a greater susceptibility to be inhibited by BCL-xL than BAX-S184A [68], and (3) a greater retention (by BCL-xL) away from OMM [68]. The differential regulation of BAX localization by BCL-xL and BCL-xLΔC is schematized in Figure 1.

S184 is one of possibly several residues phosphorylatable by survival kinases, such as AKT [84,85] or PKCζ [86]. AKT activation was shown to prevent apoptosis through the inhibition of BAX mitochondrial relocalization, in accordance with the behavior of the phosphomimetic mutant S184D, which is poorly localized to mitochondria. Interestingly, small molecules able to initiate the same conformational change induced by the dephosphorylation of Ser184 have been identified as potential BAX activators [87].

Different BAX mutants with different levels of mitochondrial localization and activity were expressed in a yeast mutant deleted for Sch9, the ortholog of AKT [88]. The deletion activated both the mitochondrial localization and the capacity of BAX mutants to release cytochrome c (Figure 2). This was in line with the expected inhibitory effect of AKT on BAX. However, the co-expression of a constitutively active mutant of AKT and BAX in yeast led to a rather different figure. AKT induced an increase of cellular BAX content that impacted the mitochondrial BAX content [82]. This effect was related, at least in part, to the phosphorylation of S184, since it did not happen with the non-phosphorylatable mutant BAX-S184V. The increased amount of mitochondrial BAX did induce an increase in the release of cytochrome c, but it was difficult to attribute this effect to an increase of the intrinsic activity (like the one observed with the phosphomimetic mutant BAX-S184D) or the increase of mitochondrial BAX content (like it was observed with BAX-S184V). It should be noted that if AKT actually phosphorylated BAX on Ser184 [82], the hypothesis that other Ser residues are also phosphorylated by this kinase cannot be ruled out. Indeed, some substitutions on Ser or Thr residues of BAX seemed to induce a loss or a gain of activity when expressed alone in yeast, as compared to wild-type BAX (Figure 3).

### 2.2. Translocation and Retrotranslocation of BAX

According to the classical view of BAX activation, the protein is retained in the cytosol in non-apoptotic cells, and is translocated to the OMM following an apoptotic signal. The demonstration that BAX could interact with anti-apoptotic proteins BCL-2 and BCL-xL on one hand, and with BH3-only proteins on the other hand is the basis of the two models of BAX activation, termed indirect and direct, respectively [92]. According to the indirect model, BAX and BCL-2 (or BCL-xL) form inactive complexes in the cytosol. Upon apoptosis initiation, BH3-only protein such as BAD become able to interact with BCL-2/BCL-xL thus liberating BAX from its interaction and allowing its mitochondrial translocation. According to the direct model, BH3-proteins such as tBID, PUMA, or BIM directly interact with BAX thus favoring a conformational change allowing its mitochondrial translocation. This has been demonstrated to happen for two BH3-only proteins, BIM [93,94] and PUMA [95]. Although the indirect model was largely supported by experiments in mammalian cells, the actual existence of a direct model for some BH3-only proteins, namely PUMA, was questioned [68]. The ectopic expression in yeast offered a good tool to confirm that the direct model could have actually occurred: indeed, yeast does not contain any anti-apoptotic proteins resembling BCL-2 and BCL-xL. It was shown that the co-expression or tBID [96] or PUMA [95] actually activated BAX, favoring its mitochondrial translocation, in full accordance with the direct model. Interestingly, the indirect model might also be reproduced in yeast, by co-expressing BAX with BCL-xL and adding the BH3-mimetic molecule ABT-737 to initiate growth arrest [68]. The sequential expression of full-length BAD or BAD-BH3 domain in yeast cells already expressing BAX and BCL-xL (alone or in combination) has also been tested but may require further adjustments in terms of relative levels of expression of the different proteins (Renault and Manon, unpublished data).

Yeast has been also used to evidence an unexpected effect of anti-apoptotic proteins on BAX localization in non-apoptotic cells. In mouse lymphocytes, the overexpression of BCL-2 [14] or BCL-xL [15] has been shown to stimulate the mitochondrial localization of BAX. This rather unexpected effect was confirmed in yeast where the co-expression of BCL-xL did induce a greater mitochondrial localization of BAX than the expression of BAX alone [15]. This effect became even more unexpected when it was demonstrated that BCL-xL (and BCL-2) could stimulate the retrotranslocation of BAX in mammalian cells [13]. Interestingly, it was shown that the C-terminal residues of BCL-xL were needed for stimulating the retrotranslocation of BAX [16]. We observed that BCL-xLΔC (devoid of the whole C-terminal helix) was able to stimulate BAX mitochondrial localization in yeast even more than full-length BCL-xL [15]. An indirect measurement of BAX retrotranslocation showed that BCL-xL induced BAX retrotranslocation from yeast mitochondria while BCL-xLΔC did not. It was thus concluded that both BCL-xL and BCL-xLΔC stimulated the translocation of BAX but that only BCL-xL also induced the retrotranslocation, thus resulting in a higher mitochondrial content when BCL-xLΔC was present [38].

### 2.3. BCL-xL-Dependent Stimulation of Translocation/Retrotranslocation v/s BAX Phosphorylation

Since BCL-xL can stimulate both BAX translocation and retrotranslocation, it is likely that an additional regulation event is involved so that the whole process is not a futile cycle. A first indication is given by the fact that a constitutively active mutant of BAX, carrying the mutation P168A changing the mobility of the C-terminal α-helix, cannot be retrotranslocated [15]. This indicates that as soon as BAX is engaged into a process of oligomerization (leading to the permeabilization of the OMM), it is not prone to retrotranslocation anymore. Like we have reported above, the phosphomimetic mutant BAX-S184D is more susceptible to interact with BCL-xL away from the OMM while the non-phosphorylatable mutant BAX-S184V is more susceptible to interact with BCL-xL in the OMM [68]. This suggested that BAX-S184D might be more easily retrotranslocated while BAX-S184V might be more easily translocated (by BCL-xL). This support the view of a cycle of phosphorylation/dephosphorylation associated with the cycle of translocation/retrotranslocation, with an escape towards BAX oligomerization and activation when apoptosis is initiated (Figure 4).

This model should now be tested through biochemical approaches, to understand how BAX phosphorylation status may influence its capacity to interact with BCL-xL, in soluble and membrane contexts. Both BCL-xL [97,98] and BAX [91,99] can be purified in soluble phase (in the absence of detergent or in the presence of low amounts of detergents or surfactants) or in membrane phases (liposomes, nanodiscs). Reconstitution assays can be combined with the use of mutants and enzymatic and non-enzymatic post-translational modifications to reproduce the different status of the proteins. For example, it has been shown that in addition to phosphorylation [20,100], BCL-xL can also be submitted to monodeamidation [21], thus expanding the number of post-translational modifications susceptible to modulate the interactions between BAX and BCL-xL. A variety of protein/protein interaction methods can be used to test co-immunoprecipitation, proton/deuterium exchange coupled to MS, Fluorescence transfer methods, DEER, etc... The critical point is to obtain purified BAX and BCL-xL under a conformation as close as possible as the native proteins: it is particularly important to verify that they are produced as intact full-length proteins and that tags and additives (detergent, surfactants, bound probes) do not change their behavior.

## 3. Regulators of BCL-2 Family Members Localization

### 3.1. Mitochondrial Receptors

Most mitochondrial proteins are encoded by nuclear DNA and thus imported into mitochondria through an intricate network of import complexes [101] (for review). The TOM (Tranlocase of Outer Membrane) and TIM (Translocase of Inner Membrane) complexes are required for protein transport through mitochondrial membranes, sorting into the adequate compartment, and assembly of mitochondrial complexes. The SAM (Sorting and Assembly Machinery) complex is required for the adequate localization of many (but not all) OMM proteins (including TOM components). The question of the involvement of any of these complexes in the mitochondrial localization of BCL-2 family members has been raised as soon as 2002 with evidence, obtained in yeast, that the receptor Tom20 could be involved in the mitochondrial localization of BCL-2 but that the process did not involve the general TOM-dependent pathway [26]. This observation was recently confirmed and extended both in yeast and human cancer cells, by showing that a peptide corresponding to the Tom20 domain interacting with BCL-2 was able to prevent the mitochondrial localization of BCL-2 [25]. It was also shown that the localization of BAX involved another receptor, Tom22, both in mammalian cells and yeast [27]. A similar conclusion has been drawn from observations on the expression of mammalian BAX in Drosophila [102]. Here again, the general TOM-dependent pathway was not involved. In fact, Tom22 was not absolutely required for BAX mitochondrial localization, since constitutively mitochondria-localized BAX (such as the ΔS184 mutant) could serve as a receptor to cytosolic wild-type BAX [28]. The conclusion was debated by another study [103] but, most mechanistic experiments of this paper were done with the BAX-S184V mutant, which has the same “receptor-like” behavior as the membrane-constitutive ΔS184 mutant, thus bypassing Tom22 requirement [28]. Ectopic expression in yeast suggested that Tom22 was however needed for the further process of BAX activation to occur: indeed, yeast cells where Tom22 was down-regulated, or incubated in the presence of a competition peptide decreasing the interaction between BAX and Tom22, did accumulate mitochondrial BAX but the protein was barely able to self-assemble into a fully active pore and rather formed inactive dimers [29]. BAX close homolog BAK has been shown to interact with metaxins (SAM components) on the OMM and VDAC2 on the inner membrane: it has been suggested that the binding of BAK with VDAC2 or metaxin 2 maintained BAK under an inactive form, while the interaction with metaxin 1 was part of BAK activation pathway [31]. Two additional papers have suggested that TOM70 was involved in the mitochondrial localization of MCL-1 [33] and that BIM could interact with both TOM70 and TOM20 [30] without, however, any obvious effect on its mitochondrial localization. To date, no putative receptor for BCL-xL has been identified but, as reported above, the sequences surrounding the C-terminal domain of BCL-xL are clearly different from those of BCL-2 and BAX [74], possibly accounting for the peculiar behavior of this protein.

### 3.2. Mitochondria-Associated Membranes (MAM) and Mitochondria-ER Contacts (MERC)

MERC are defined as close contacts between OMM and domains of the ER [22] (for review). These ER domains, which have a proteic and lipid composition distinct from the bulk ER have been called MAM [104] stable structure and their number and the nature of their components may change with the physiological status of the cell, reflecting the range of their possible functions. A proteomic analysis involving a proximity-driven biotinylation of proteins surrounding a linker protein between OMM and ER has been reported [23]. The K21 of BAX was found among biotinylated residues near the OMM bait, Tom20 but no other BCL-2 family member was found in the assay (note, however, that cells were not committed to any form of apoptosis). This strongly suggested that in non-apoptotic cells, BAX might be located, at least in part, in MERC (that should be understood as including both the OMM and ER membranes surrounding the contacts).

This was in line with two previous observations indicating that MAM stability could regulate BAX mitochondrial localization. Co-expression of anti-apoptotic viral protein vMIA (Viral Mitochondria-localized Inhibitor of Apoptosis) was shown to induce the accumulation of BAX in MAM, which was followed by a proteosomal degradation of the protein [105]. In another study, the localization of BAX in yeast cells carrying a deletion of a gene encoding the protein Mdm34 (a component of the ERMES complex for ER Mitochondria Encounter Structure, involved in MERC stability), was moderately but significantly affected, with consequences on the capacity to release cytochrome c [24]. It can be hypothesized that BAX transited through the ER and MAM before to reach OMM, in a process requiring MERC stability. The destabilization of these contacts generated by the deletion of Mdm34 might impair BAX transit. Interestingly, ERMES components are also involved in yeast physiological programmed cell death induced by acetic acid [106].

Can this observation be extended to mammalian cells? Opposite to yeast, where the requirement of ERMES complex has been established [107,108] (for reviews), the stability of MAM and MERC in mammalian cells is rather dependent on several complexes, depending on conditions and of the function under study: PS transport, calcium transport, mitochondria dynamics, mitophagy, etc... [104,109] (for reviews). A common feature of MAM stability in yeast and mammals is the regulation by the GTPase Miro1/2 in mammals [110], in plants [111] or Gem1 in yeast [112]. The consequences of the activity of Miro1/2/Gem1 on BAX mitochondrial localization should then be investigated.

It is possible to correlate the role of MAM stability on BAX transit to the process of translocation/retrotranslocation. Indeed, it is more energetically favorable for BAX to shuttle back and forth between two membranes (ER and OMM) than between a membrane phase and a soluble phase (cytosol and OMM). The hypothetic regulation illustrated in Figure 4 can then be completed by assuming that the translocation/retrotranslocation process occurs at ER/mitochondria contact sites (Figure 5). Separate measurement of BAX translocation and retrotranslocation (by using the differential capacity of BCL-xL and BCL-xLΔC to stimulate these two processes) could then be measured under conditions where MAM are destabilized (in a mutant of the ERMES complex in Yeast, or of the complexes involved in MAM stability in Mammals) or, conversely, stabilized with chimeric proteins linking both membranes [113,114].

## 4. Concluding Remarks

The unusual regulation of BAX localization remains a major question to understand how the apoptotic process is induced (or not) following an apoptotic stimulus such as the one triggered by cancer therapies. This question is directly related to several peculiar properties of cancer cells. For example, it is well established that most cancer cells overexpress anti-apoptotic proteins BCL-2, BCL-xL or else MCL-1. However, they also often overexpress pro-apoptotic proteins BAX or BAK: for example, BAX was reported to be overexpressed (alone or in combination with BCL-2) in about one third of breast cancers [116] or in several pancreatic cancer cell lines [117]. The observations that both BCL-2 and BCL-xL overexpression induced an increase of the mitochondrial localization of BAX was rather unexpected, particularly if one considers that this observation was made at about the same time that the process of BAX retrotranslocation by BCL-xL was reported. In the absence of regulations, this appears as a futile cycle, irrelevant to the implementation of apoptosis. However, if one considers the possible regulations reported above, this appears as a sophisticated process allowing the balancing of the amount and level of activity of BAX in the OMM. Of course, many studies are still required to refine and ultimately confirm or infirm this model of regulation.

It is now well established that BH3-mimetic molecules that decrease the interaction between BAX and BCL-2/BCL-xL not only allow the release of BAX from this inhibiting interaction [118] (for review), but released BAX under a primed conformation ready for insertion and permeabilization of the OMM [15,115]. However, as therapeutic agents, these molecules are not without disadvantages: BCL-xL is required for platelets survival and its unregulated inhibition is a cause of thrombocytopenia [119]. ABT-199 (venetoclax) has been designed to be selective to BCL-2 and thus does not cause thrombocytopenia [120]. However, major adverse effects are still present, implying that this type of treatment requires a careful selection of patients [121]. It might therefore be desirable to identify molecules that would stimulate the transfer of BAX from the ER to the OMM, independently from the phosphorylation status of BAX and/or from its interaction with BCL-2/BCL-xL. For example, one can assume that stabilizing MERC might help the mitochondrial translocation of BAX in cells committed to apoptosis. This might bypass the phosphorylation status of BAX and the interaction with BCL-xL to increase BAX localization at the OMM. Considering that most cancer cells display a constitutively active level of AKT activity [122,123] (for reviews) and that BAX is one of the multiple targets of this kinase [84,85], combining the action of BH3 mimetics with the stimulation of BAX mitochondrial localization even in the unfavorable context of AKT activation might help to refine therapeutic treatments.

## Figures and Tables

**Figure 1 ijms-22-04086-f001:**
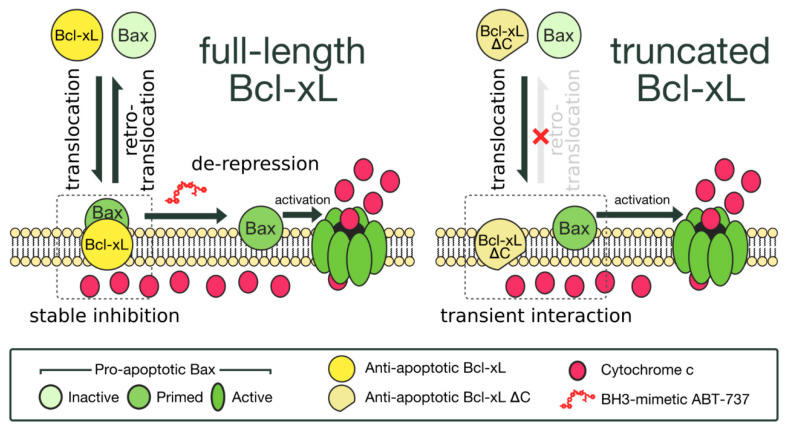
Schematic representation of the differential effect of BCL-xL on BAX dynamics (reproduced from [38]). **Left**: BAX is under a dynamic equilibrium between mitochondria and cytosol, owing to the processes of BCL-xL-driven translocation and retrotranslocation. Mitochondrial BAX is primed but remains inactive because of the interaction with BCL-xL. The addition of a BH3-mimetic (or the expression of a BH3-only partner) enables the derepression of the interaction and the full activation of BAX. **Right**: BCL-xLΔC stimulates BAX translocation but not BAX retrotranslocation, leading to a higher mitochondrial BAX content. Furthermore, the interaction between BAX and BCL-xLΔC is weaker than with BCL-xL, and BAX can therefore be activated in the absence of BH3-only protein or BH3 mimetics.

**Figure 2 ijms-22-04086-f002:**
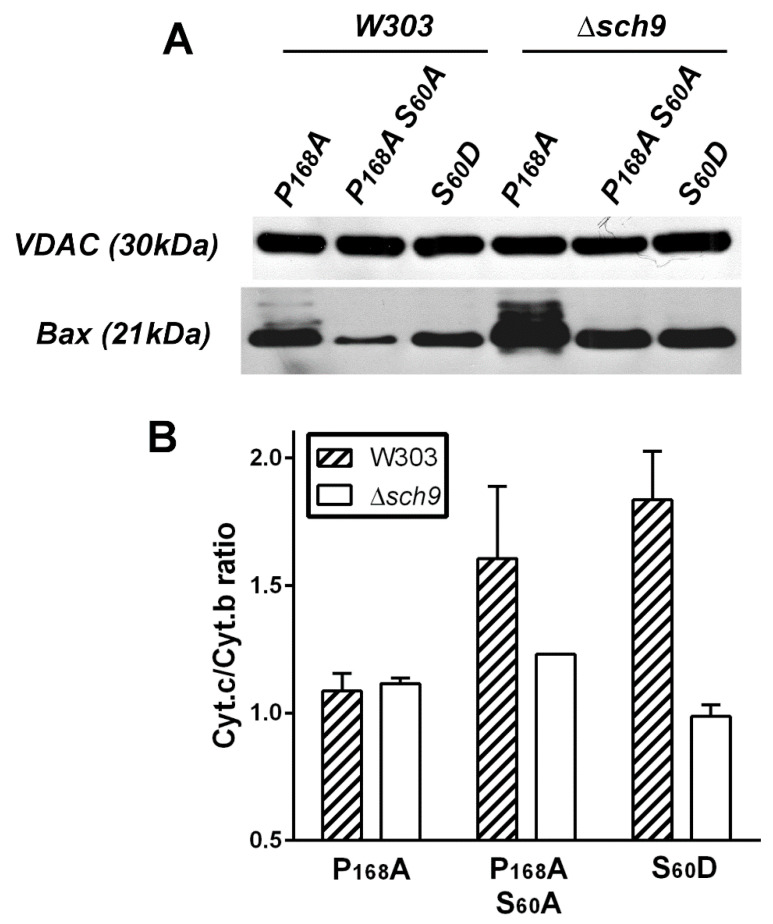
Stimulation of BAX mitochondrial localization and activation in Sch9-deleted yeast. Different BAX mutants were expressed in the wild-type strain W303 or in a mutant carrying a deletion of the gene of the AKT-ortholog Sch9. BAX content was measured in isolated mitochondria with OMM protein VDAC (Voltage Dependent Anion Channel) as a loading control. As previously reported, the cellular content of the three mutants is about the same but they exhibit different mitochondrial localization and level of activity (see Figures 1 and 5 in ref. [81]). The mutant P168A has both a high activity and high mitochondrial localization, and the mutants P168A/S60A and S60D have a weak activity with a weak and intermediate mitochondrial localization, respectively [81]. The P168 residue is in the short loop between α8 and α9 helices and its substitution was shown to stimulate BAX in mammalian cells [89], yeast cells [90], and in vitro [91] by favoring the movement of the α9 helix. The S60 residue is part of a consensus sequence KKLS that might be targeted by a Protein Kinase A, and is in the α2 helix in close proximity to a negatively charged residue D33 located in the α1 helix (see ref. [81] for details). (**A**) Western-Blots showing the presence of BAX in isolated mitochondria. (**B**) Redox spectrophotometry measurements of Cyt.c and Cyt.b in mitochondria. Mitochondrial cytochromes absorb light differently in their oxidized and reduced states. The maximal difference between chemically reduced (with Sodium Dithionite) and oxidized (with Potassium Ferricyanide) cytochromes is at 550 nm for cytochrome c and 561nm for cytochrome b with isosbestic points at 540 and 575 nm., with Δε (reduced *minus* oxidized) = 18,000 M^−1^·cm^−1^ for both cytochromes, allowing the precise calculation of their remaining concentration in mitochondria. Cytochrome b is part of the inner membrane respiratory Complex III and is not released during OMM permeabilization, thus serving as an internal control. The decrease of the ratio mitochondrial Cyt.c/Cyt.b ratio reflects the capacity of BAX to release Cyt.c [15,81].

**Figure 3 ijms-22-04086-f003:**
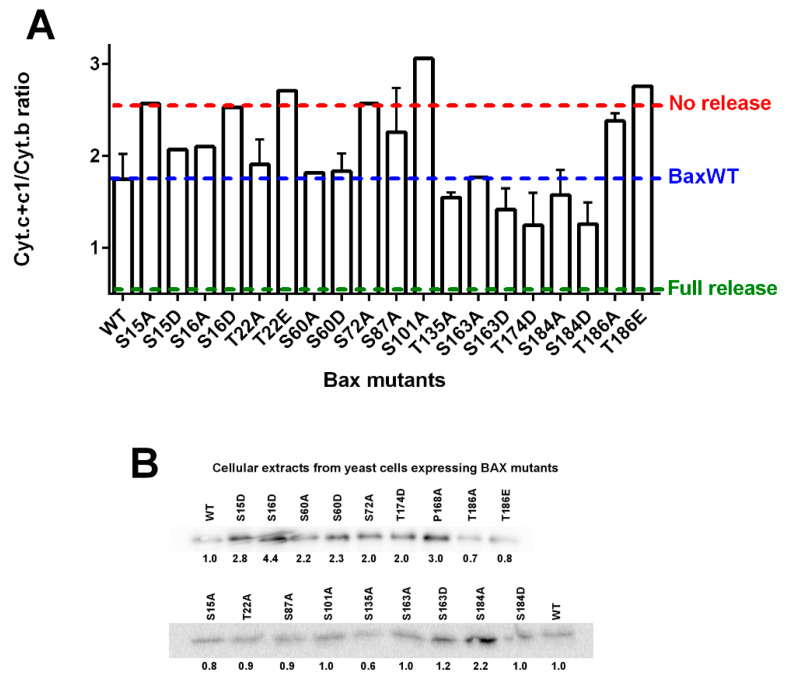
Effect of substitutions of potentially phosphorylatable residues on BAX activity. Different mutants of BAX carrying substitutions of potentially phosphorylatable residues have been expressed in yeast [80]. Potentially phosphorylatable residues were selected based on a mass spectrometry analysis of purified BAX incubated in the presence of AKT or GSK3β, or known data from the literature. (**A**) After mitochondria isolation, the mitochondrial ratio Cyt.c+c1/Cyt.b was measured by redox spectrophotometry [15]. The blue dashed line indicates the ratio measured on wild-type BAX. The red dashed line indicates the average value measured in W303 yeast strain not expressing BAX. The green dashed line indicated the theoretical ratio value if all cytochrome was released (corresponding to the ratio Cyt.c1/Cyt.b of 0.5 in Respiratory Complex III). Some mutants are significantly more active than wild-type BAX, such as S184D that has been characterized extensively [81,82], but some mutants also appeared to be less active than wild-type BAX. The level of expression of each BAX mutant might be different (**B**), but there is no obvious correlation between the content and the capacity to release cytochrome c: for example, the mutant BAX-S16D (inactive) is expressed at about two-times the level of BAX-WT, the mutant BAX-S135A (moderately active) is expressed at about half the level of BAX-WT and the mutant BAX-S184D (very active) is expressed at about the same level as BAX-WT.

**Figure 4 ijms-22-04086-f004:**
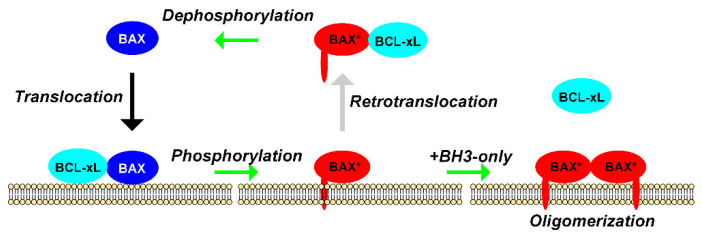
Schematic representation of the link between BAX phosphorylation and Translocation. Phosphorylated BAX (by AKT and other survival kinases) is maintained away from mitochondria while non-phosphorylated BAX is in the OMM, both in yeast and mammalian cells [80,83]. The interaction of BAX with BCL-xL stimulate both BAX translocation [15] and BAX retrotranslocation [13,15]. Heterodimers between phosphomimetic BAX and BCL-xL are more present away from mitochondria while heterodimers between non-phosphorylatable BAX and BCL-xL are more present at the OMM [68]. The release of the interaction between mitochondrial BAX and BCL-xL by BH3-only proteins (or BH3 mimetics) triggers the oligomerization/activation process of BAX [15].

**Figure 5 ijms-22-04086-f005:**
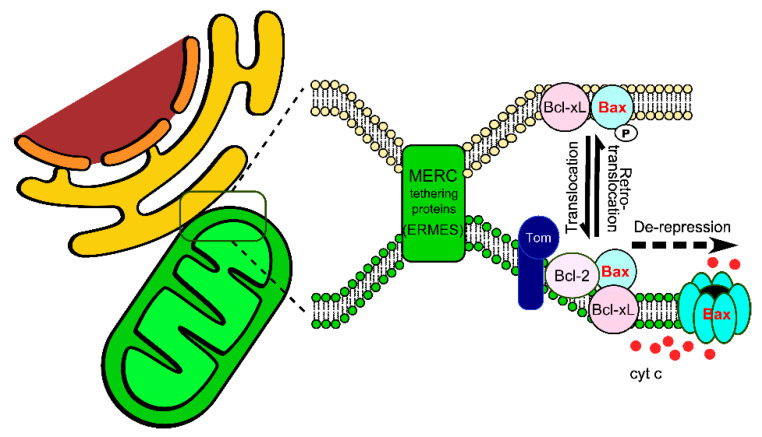
Schematic representation of the involvement of Mitochondria-ER contact sites (MERC) in BAX dynamics. Translocation and retrotranslocation processes are likely favored at MERC. The forced destabilization of the tethering complex ERMES in yeast led to a lower capacity of BAX to localize to the OMM and to promote the release of cytochrome c [24]. Components of the TOM complex Tom20 and Tom22 help the localization and/or the adequate conformation of BCL-2 [25,26] and BAX [27,28,29], respectively. Derepression by BH3-only proteins or BH3 mimetics allows the release of BAX under its active conformation able to form the pore [15,115].

## Data Availability

Data and material are available on request.

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
