# Peer review of "Contribution of Yeast Studies to the Understanding of BCL-2 Family Intracellular Trafficking"

_ijms, 2021, doi:10.3390/ijms22084086_

Round 1

Reviewer 1 Report

Eyitayo et al. provide a very useful review of the contributions of studies in yeast towards understanding the mechanisms governing subcellular localizations of mammalian BCL2 family proteins. Overall, the information is accurate and assimilates a large body of information into logical conclusions. There are a number of places that need clarification or reordering to improve overall comprehension and flow, and these are listed below to increase impact and for a broader audience. While these are relatively minor points with minor fixes, they are critical for understanding.

  1. The Abstract assumes a level of knowledge by readers who may not be intimately familiar with the field. Suggested modifications:
    1. …ectopic expression of mammalian BCL2 family proteins in yeast Sc, which lack BCL2 homologs, …
    2. …takes advantage of the powerful tools available in yeast to probe … that are largely conserved between yeast and mammals.”
    3. Because this is a review, plus new unpublished results, the closing sentence of the Abstract should provide both a summary of salient contributions of yeast described in this review or a summary of the model being put forth in Fig. 4, as well as a statement about the new findings – along the line: … here we review…/suggest a model where … , and provide new evidence for such…
  2. Before discussing the functions of BCL-2 family proteins in regulating apoptotic cell death, a topic not covered in this review, the opening paragraph of the Introduction is an important place to elaborate on how yeast can facilitate BCL-2 family studies by expanding on this point raised in the Abstract (and not wait until paragraph 4).
  3. Please insert citations to support the general overview of BCL-2 family proteins in regulating apoptosis in the introductory paragraph: “BCL-2 family members…”.
  4. The next paragraph ending: BCL-2 and BAX localization “is also multiple and regulated.” – meaning multiple subcellular localizations? Functions? Other? Clarify if the Bcl-2 family functions being studied in yeast are those related to cell death, or only non-death functions.
  5. The definitions of these terms need elaboration: MERC (a place, protein complex, includes two membranes – mitochondrial and ER, also includes the proteins that tether them, or only the proteins that tether these membranes?). Same applies to MAM, and ERMES. Include a clear explanation of the differences between MERC and ERMES. Explain here and/or in section 2.2.
  6. In many places, the authors refer to “stability” without sufficient cause, or more explanation is needed. For example, “This suggests that the stability of the … MERCs could be an element of the regulation of Bax localization.” If indeed the ‘stabilization’ of a protein-membrane/membrane-membrane complex is being discussed, what is the evidence that this impinges on BAX?
  7. The authors should clarify their conclusions regarding the Ybh/Bxi1 yeast protein. Is there a known function of the Bh3 domain in Bxi1? Clarify what is known/not known.
  8. The paragraph starting on line 69 is important history to explain how yeast came to be used for studying BCL-2 family proteins. However, many will not grasp the important points without more help. Those who haven’t performed a yeast two-hybrid will not know what is a “fusion protein”, and the authors do not explain the concept that BCL-2 is used as a bait to identify mammalian binding partners by co-expressing human cDNA libraries in yeast, affording the opportunity to observe the unexpected finding that BCL-2 proteins alone can protect yeast from cell death.
  9. In addition, there is an even earlier paper by Dale Bredesen that should be cited for demonstrating that BCL-2 inhibits yeast cell death (Kane et al. Science 1993).
  10. Regarding the discussion about non-canonical BH3-like proteins:
    1. The discussion of non-canonical BH3-like proteins was not sufficiently transparent to be certain what is the intended message.
    2. Also, the authors may wish to back off on Bxi1/Ybh3 to improve the article’s credibility with the majority opinion in the cell death field.
    3. There is surely little doubt that endogenous BCL-2/BCLxL interacts with endogenous Beclin 1 to alter function, are the authors contesting this in some way? Please explain.
    4. Statements made about the example BH-3 proteins (line 97) do not appear to be supported by the statements provided.
    5. BCL2L13/Rambo is not a legal “ortholog” of Atg32 as stated, which would also be inconsistent with earlier statements in this review.
  11. The language in the first paragraph of section 1.1 is not clear and not consistent with later statements. [“needed” is generally interpreted as necessary; tailless BCL2 proteins can be recruited to mitochondria by other means at least in mammalian cells; what is the meaning of “in any case”?].
  12. The paragraph beginning line 121 is also somewhat vague; what is the meaning of “cleaner”; “ubiquitous” is misused in the following paragraph.
  13. At several points in the review, one wonders what is the effect of BAX and BAX mutants on yeast cell death, but this is not mentioned. Though perhaps beyond the scope of this review, it is still important for interpreting some of the results discussed – for example does the death of yeast cells impact the “stability” of expressed BAX(S184D)? Is yeast BAX activated (e.g. detected with 6A7 antibody)?
  14. Paragraph beginning line 188 – is this a discussion about yeast or mammalian cells? Please check and modify the start of all other paragraphs to avoid confusing readers.
  15. Same paragraph – replace “All things considered” with, “Taking all the evidence into account”, if this was the intention. However, the evidence supporting point #2 was not found, or the wording does not convey the intended message – what function of BAX(S184D) is being inhibited by BCLxL – cell death? or this refers only to point #3?
  16. Figure 1 contains two panels as explained in the legend, but the panels are not labeled in the figure, and not cited as Fig. 1A and B in the text.
  17. Some discussion about phosphorylation should precede the discussion of mutations of Ser 184 of BAX to mimic unphosphorylated vs. phosphorylated BAX to first justify the lengthy discussion of BAX S184A vs. S184D.
  18. If correct, the authors want to state that increased mito localization of S184D, and decreased localization of S184A, is because BCL-xLΔC is still able to localize S184A and S184D to the membrane, but is no longer able to retrotranslocate S184D out of the mitochondrial membrane. Therefore, full-length BCLxL is able to regulate BAX translocation and retrotranslocation. This is cumbersome to read and additional synthetic statements would be helpful to fully grasp the intended meanings.
  19. Does the amount of mitochondrial BAX shown in the immunoblots in Fig. 1 reflect the amount of BAX translocated to mitochondria or the amount of total BAX in the cells? Without knowing this answer, conclusions are limited, though the finding in general is interesting. Similarly, are the P168A/S60A mutants lowly expressed due to BAX-induced cell death or reduced translocation to mitochondria without the effects of Akt/Sch9 function? Please justify the selection of P168 and S60 for mutation should be provided. The findings could be better explained to have greater impact. Including citations in Fig 1 legend seems to imply these findings were previously reported – please clarify in the text if these are new data.
  20. Similarly, without immunoblots for the data in Fig. 2, the observed effects of this impressive set of BAX mutants on yeast cytochrome c release are challenging to interpret.
  21. Please briefly explain the concept of the Cyt c/Cyt b ratio assay in the text to convince readers.
  22. The authors explain that the direct BAX activation mechanism also works in yeast, but was the indirect mechanism ever tested in yeast? If not, please state such.
  23. In the Conclusion section, provide a few more details about the proposed model in Figure 4 – how in detail would their proposed model be applied to understanding cancer mechanisms?
  24. References [83] and [84] are not found in the Reference section.
  25. Upper case lettering for mammalian proteins (BCL-2).

Thank you for the interesting article.

Author Response

Reviewer 1

Comments and Suggestions for Authors

Eyitayo et al. provide a very useful review of the contributions of studies in yeast towards understanding the mechanisms governing subcellular localizations of mammalian BCL2 family proteins. Overall, the information is accurate and assimilates a large body of information into logical conclusions. There are a number of places that need clarification or reordering to improve overall comprehension and flow, and these are listed below to increase impact and for a broader audience. While these are relatively minor points with minor fixes, they are critical for understanding.

1. The Abstract assumes a level of knowledge by readers who may not be intimately familiar with the field. Suggested modifications:

1. …ectopic expression of mammalian BCL2 family proteins in yeast Sc, which lack BCL2 homologs, …

2. …takes advantage of the powerful tools available in yeast to probe … that are largely conserved between yeast and mammals.”

3. Because this is a review, plus new unpublished results, the closing sentence of the Abstract should provide both a summary of salient contributions of yeast described in this review or a summary of the model being put forth in Fig. 4, as well as a statement about the new findings – along the line: … here we review…/suggest a model where … , and provide new evidence for such…

We have included the suggestions of the reviewer.

2. Before discussing the functions of BCL-2 family proteins in regulating apoptotic cell death, a topic not covered in this review, the opening paragraph of the Introduction is an important place to elaborate on how yeast can facilitate BCL-2 family studies by expanding on this point raised in the Abstract (and not wait until paragraph 4).

We have included some sentences in the firs paragraph to emphasize the interest of yeast for these type of studies.

3. Please insert citations to support the general overview of BCL-2 family proteins in regulating apoptosis in the introductory paragraph: “BCL-2 family members…”.

We have considerably rewritten this part and we have included relevant references, both to original papers and to more recent reviews.

4. The next paragraph ending: BCL-2 and BAX localization “is also multiple and regulated.” – meaning multiple subcellular localizations? Functions? Other? Clarify if the Bcl-2 family functions being studied in yeast are those related to cell death, or only non-death functions.

Our initial text was somewhat imprecise. We have added some sentences to take into account the “non-apoptotic” functions of BCL-2 family members.

5. The definitions of these terms need elaboration: MERC (a place, protein complex, includes two membranes – mitochondrial and ER, also includes the proteins that tether them, or only the proteins that tether these membranes?). Same applies to MAM, and ERMES. Include a clear explanation of the differences between MERC and ERMES. Explain here and/or in section 2.2.

We have given the definitions of these terms as soon as they are cited in the text. Although the term “MERC” (that defines the whole contact zone) is less widely known than the terms “MAM” (ER domains in contact with mitochondria) and “ERMES” (the yeast complex involved in these contacts), we found it was appropriate.

6. In many places, the authors refer to “stability” without sufficient cause, or more explanation is needed. For example, “This suggests that the stability of the … MERCs could be an element of the regulation of Bax localization.” If indeed the ‘stabilization’ of a protein-membrane/membrane-membrane complex is being discussed, what is the evidence that this impinges on BAX?

We found that the term “stability” (applied to MAM and MERC) includes a notion of dynamics, suggesting that contact sites can appear when followed for some time or depending on the physiological status of the cell. Maybe our wording of how this stability can contribute to BAX behaviour was confusing. We rewrote several sentences for more clarity.

7. The authors should clarify their conclusions regarding the Ybh/Bxi1 yeast protein. Is there a known function of the Bh3 domain in Bxi1? Clarify what is known/not known.

We have added some discussion about Ybh3/Bxi1 and actually corrected several points that were imprecise in our first version.

8. The paragraph starting on line 69 is important history to explain how yeast came to be used for studying BCL-2 family proteins. However, many will not grasp the important points without more help. Those who haven’t performed a yeast two-hybrid will not know what is a “fusion protein”, and the authors do not explain the concept that BCL-2 is used as a bait to identify mammalian binding partners by co-expressing human cDNA libraries in yeast, affording the opportunity to observe the unexpected finding that BCL-2 proteins alone can protect yeast from cell death.

We have included several sentences to better explain the different ways of how yeast can contribute to the knowledge about cell death mechanisms.

9. In addition, there is an even earlier paper by Dale Bredesen that should be cited for demonstrating that BCL-2 inhibits yeast cell death (Kane et al. Science 1993).

We have included this reference and also included a later paper by Polcic's group about a somilar study with BCL-xL.

10. Regarding the discussion about non-canonical BH3-like proteins:

1. The discussion of non-canonical BH3-like proteins was not sufficiently transparent to be certain what is the intended message.

2. Also, the authors may wish to back off on Bxi1/Ybh3 to improve the article’s credibility with the majority opinion in the cell death field.

3. There is surely little doubt that endogenous BCL-2/BCLxL interacts with endogenous Beclin 1 to alter function, are the authors contesting this in some way? Please explain.

4. Statements made about the example BH-3 proteins (line 97) do not appear to be supported by the statements provided.

5. BCL2L13/Rambo is not a legal “ortholog” of Atg32 as stated, which would also be inconsistent with earlier statements in this review.

We have completely rewrote this paragraph, with a better distinction between the “classical” model, based on primary structure/functions of these proteins and the “evolutionary” model, based on phylogenetic analyses. We think that both views are important.

11. The language in the first paragraph of section 1.1 is not clear and not consistent with later statements. [“needed” is generally interpreted as necessary; tailless BCL2 proteins can be recruited to mitochondria by other means at least in mammalian cells; what is the meaning of “in any case”?].

We have added a sentence to better explain the difference between Bcl-2/Bcl-xL behavior and Bax behavior.

12. The paragraph beginning line 121 is also somewhat vague; what is the meaning of “cleaner”; “ubiquitous” is misused in the following paragraph.

We have modified the sentence, to state that the mutant A221R is even “less mitochondrial” than the deleted protein (both in yeast and mammals).

13. At several points in the review, one wonders what is the effect of BAX and BAX mutants on yeast cell death, but this is not mentioned. Though perhaps beyond the scope of this review, it is still important for interpreting some of the results discussed – for example does the death of yeast cells impact the “stability” of expressed BAX(S184D)? Is yeast BAX activated (e.g. detected with 6A7 antibody)?

We have added se veral sentences to avoid confusions between experiments in yeast and in mammalian cells.

14. Paragraph beginning line 188 – is this a discussion about yeast or mammalian cells? Please check and modify the start of all other paragraphs to avoid confusing readers.

same as above

15. Same paragraph – replace “All things considered” with, “Taking all the evidence into account”, if this was the intention. However, the evidence supporting point #2 was not found, or the wording does not convey the intended message – what function of BAX(S184D) is being inhibited by BCLxL – cell death? or this refers only to point #3?

The confusion came from the fact that it was not clear that we compared BaxS184D with BaxS184A (not BaxWT) in the paper by Garenne et al. We corrected the sentence for more clarity.

16. Figure 1 contains two panels as explained in the legend, but the panels are not labeled in the figure, and not cited as Fig. 1A and B in the text.

We have corrected and added more information in the legend of the Figure

17. Some discussion about phosphorylation should precede the discussion of mutations of Ser 184 of BAX to mimic unphosphorylated vs. phosphorylated BAX to first justify the lengthy discussion of BAX S184A vs. S184D.

This was a deliberate choice to discuss the behavior of the S184 mutants (on the basis of the initial description of the deltaS184 mutant by Richard Youle) before to discuss the fact that this residue can be phosphorylated, because the actual effect of phosphorylation is not exactly that was expected from the substituted mutants.

18. If correct, the authors want to state that increased mito localization of S184D, and decreased localization of S184A, is because BCL-xLΔC is still able to localize S184A and S184D to the membrane, but is no longer able to retrotranslocate S184D out of the mitochondrial membrane. Therefore, full-length BCLxL is able to regulate BAX translocation and retrotranslocation. This is cumbersome to read and additional synthetic statements would be helpful to fully grasp the intended meanings.

We have reworded this part for a better explanation and we have included a figure that we previously published in another paper [ref 38].

19. Does the amount of mitochondrial BAX shown in the immunoblots in Fig. 1 reflect the amount of BAX translocated to mitochondria or the amount of total BAX in the cells? Without knowing this answer, conclusions are limited, though the finding in general is interesting. Similarly, are the P168A/S60A mutants lowly expressed due to BAX-induced cell death or reduced translocation to mitochondria without the effects of Akt/Sch9 function? Please justify the selection of P168 and S60 for mutation should be provided. The findings could be better explained to have greater impact. Including citations in Fig 1 legend seems to imply these findings were previously reported – please clarify in the text if these are new data.

In Fig.1 (now Fig.2), the blots show the presence of BAX in isolated mitochondria from wild-type strain (W303) expressing the mutants has been previously published [81]. The new data are the expression in the Δsch9 mutant, that was not published at the time but is from the same series of experiments. We have also included a better explanation about the choices of these mutants in the legend.

20. Similarly, without immunoblots for the data in Fig. 2, the observed effects of this impressive set of BAX mutants on yeast cytochrome c release are challenging to interpret.

We have included a blot showing the cellular levels of expression of these mutants, in comparison to wild-type Bax, showing some differences that, however, cannot explain the differences of cytochrome c release.

21. Please briefly explain the concept of the Cyt c/Cyt b ratio assay in the text to convince readers.

We have detailed the measurements in the legend of new Fig.2 (old Fig.1)

22. The authors explain that the direct BAX activation mechanism also works in yeast, but was the indirect mechanism ever tested in yeast? If not, please state such.

We have included a senetence referring to a paper [68] where a very indirect measurement suggested that the indirect mechanism also works in yeast. We have tried to do a more direct measurement by co-expressing BAD (or BAD-BH3], but these experiments would require to be refined.

23. In the Conclusion section, provide a few more details about the proposed model in Figure 4 – how in detail would their proposed model be applied to understanding cancer mechanisms?

We have extennded the discussion about the possible interest for cancer mechanisms.

24. References [83] and [84] are not found in the Reference section.

The numbering of reference has been changed and checked.

25. Upper case lettering for mammalian proteins (BCL-2).

This has been done

Thank you for the interesting article.)

We wish to thank the reviewer for helping us to improve the manuscript.

Reviewer 2 Report

The authors, who are experts in their field, give a comprehensive, well-written and structured overview of the molecular determinants that shape localization of Bax, Bcl-2 and Bcl-XL. Given the overall quality and clarity of the review, I only have some minor remarks, which I outline below.

General remarks

  1. While the review is about the canonical function of Bcl-2-proteins, I feel it is important to briefly mention and provide a reference that Bcl-2-proteins have moonlighting functions too.

  1. The focus is largely on Bcl-2, Bax and Bcl-XL, but there are other members à Why do the authors not discuss them. Is there no information available, is it beyond their scope? Please provide a reason.

  1. I feel the canonical role of the Bcl-2-protein family members can be a bit more detailed in the beginning of the introduction, e.g. mention BH domains, what they are and how the proteins interact.

  1. Figure 1 should be explained better, before it is shown. It is not clear what all the mutants do until much later in the review. So Figure 1 should be more extensively discussed in the text and the legend should be clarified. Additionally, it would be good if the authors could show the purity of the mitochondrial fraction and similar expression levels of Bax mutants in the whole-cell lysates.

Specific remarks

R40: Please make the difference between MERCs and MAMs clear here. It is explained later, but it would fit better here.

R51: contextualize metaxins and why they are relevant for this part of the review. They are explained later, but it would be good to explain them already here, since it is not clear.

R64: Maybe write a sentence on what BH3 domains are and why they are important in the introduction

R64-68: The relevance of this part is not clear at this point

R88: I think the more classic way of categorizing the Bcl-2-protein family members is first anti-apoptotic which have 4 BH domains vs. anti-apoptotic which can be divided in effectors (Bax/Bak/Bok) with multiple BH domains and the BH3-only proteins. The last category is then subdivided in activators and sensitizers. It is more clear to describe them like this, compared to including Bim in the first category.

If the authors retain this structural basis, it may be good to discuss what the structure of Bcl-2 and the others looks like.

R99: The referral to overexpression here, suggests that these interactions do not happen in normal physiological conditions. Maybe the authors could scrap the sentence or elaborate on this.

R97: Maybe briefly contextualize Nix and Beclin

R105: Can the authors give a reference for the statement: “Most proteins having a non-canonical BH3 domains are also inserted in OMM”? The IP3R is also found to have BH-like domains but is demonstrably in the ER.

In section 1.1 it is worth mentioning and refering that both Bcl-2 and Bcl-XL have been found at the ER as well. Maybe this is also a good place to mention that Bcl-2a/-b isoforms exist, the B-isoform without its transmembrane and that this may impact localization or interaction with other proteins PMID 27494888. I do not know whether other isoforms exist of for example Bcl-XL but it is worth mentioning.

R135-137: The authors should explain how the localization is modulated: more mitochondrial or no?

Author Response

Comments and Suggestions for Authors

The authors, who are experts in their field, give a comprehensive, well-written and structured overview of the molecular determinants that shape localization of Bax, Bcl-2 and Bcl-XL. Given the overall quality and clarity of the review, I only have some minor remarks, which I outline below.

General remarks

  1. While the review is about the canonical function of Bcl-2-proteins, I feel it is important to briefly mention and provide a reference that Bcl-2-proteins have moonlighting functions too.

We have included information about the other roles of BCL-2 family members, namely in Ca2+ regulation

  1. The focus is largely on Bcl-2, Bax and Bcl-XL, but there are other members à Why do the authors not discuss them. Is there no information available, is it beyond their scope? Please provide a reason.

BAK and MCL-1 have not been largely studied in yeast. The reason for MCL-1 is technical: the protein is very instable and very rapidly degraded in yeast, making it difficult to do biochemical studies. Some experiments have been done with BAK (including by us) but there is not enough relevant data available to help understanding its regulation. Data in mammalian cells are much more reliable.

  1. I feel the canonical role of the Bcl-2-protein family members can be a bit more detailed in the beginning of the introduction, e.g. mention BH domains, what they are and how the proteins interact.

 We have rewritten this part.

  1. Figure 1 should be explained better, before it is shown. It is not clear what all the mutants do until much later in the review. So Figure 1 should be more extensively discussed in the text and the legend should be clarified. Additionally, it would be good if the authors could show the purity of the mitochondrial fraction and similar expression levels of Bax mutants in the whole-cell lysates.

    As also answered to a similar comment by reviewer 1, the “wild-type” part of this Figure has already been published, with all the required controls [81]. The new data are about the Δsch9 mutant. The legend has been clarified (now Figure 2).

Specific remarks

R40: Please make the difference between MERCs and MAMs clear here. It is explained later, but it would fit better here.

This has been done

R51: contextualize metaxins and why they are relevant for this part of the review. They are explained later, but it would be good to explain them already here, since it is not clear.

This has been precised.

R64: Maybe write a sentence on what BH3 domains are and why they are important in the introduction

According a similar comment by reviewer 1, we have explained in more details the concepts of “canonical” and “non-canonical” BH3 domains

R64-68: The relevance of this part is not clear at this point

We hae removed the sentences and re-explaned later

R88: I think the more classic way of categorizing the Bcl-2-protein family members is first anti-apoptotic which have 4 BH domains vs. anti-apoptotic which can be divided in effectors (Bax/Bak/Bok) with multiple BH domains and the BH3-only proteins. The last category is then subdivided in activators and sensitizers. It is more clear to describe them like this, compared to including Bim in the first category.

If the authors retain this structural basis, it may be good to discuss what the structure of Bcl-2 and the others looks like.

The discussionhere has been extended to better explain the different ways tyo classify BCL-2 family members

R99: The referral to overexpression here, suggests that these interactions do not happen in normal physiological conditions. Maybe the authors could scrap the sentence or elaborate on this.

Since this part has been rewritten, this sentence has been removed.

R97: Maybe briefly contextualize Nix and Beclin

This part has been rewritten and Beclin, Nix and Rambo have been more detailed

R105: Can the authors give a reference for the statement: “Most proteins having a non-canonical BH3 domains are also inserted in OMM”? The IP3R is also found to have BH-like domains but is demonstrably in the ER.

This was for Nix and BCL2L13/Rambo. We reformulated the sentence to take into account this information.

In section 1.1 it is worth mentioning and refering that both Bcl-2 and Bcl-XL have been found at the ER as well. Maybe this is also a good place to mention that Bcl-2a/-b isoforms exist, the B-isoform without its transmembrane and that this may impact localization or interaction with other proteins PMID 27494888. I do not know whether other isoforms exist of for example Bcl-XL but it is worth mentioning.

A sentence and a reference to a relevant review by Popgeorgiev et al. [18] has been included. We also added a reference to Bcl-2β.

R135-137: The authors should explain how the localization is modulated: more mitochondrial or no?

We have replaced “modulated” by “increased”

Reviewer 3 Report

Eyitayo et al. reviewed the role of the BCL-2 family in the context of intracellular trafficking.  This is a comprehensive review for general readers.  There are some minor issues to be addressed.

  • In general, studies in yeast are confirmatory with the studies in mammalian cell. Thus, the title is not quite reflective of what the authors mainly describe in the text.
  • The strain and mutants in Figure 1 are unclear. Please describe in detail.
  • What is “ERMES” (Line 367) abbreviated?

Author Response

Eyitayo et al. reviewed the role of the BCL-2 family in the context of intracellular trafficking. This is a comprehensive review for general readers. There are some minor issues to be addressed.

In general, studies in yeast are confirmatory with the studies in mammalian cell. Thus, the title is not quite reflective of what the authors mainly describe in the text.

We changed the title to “downplay” yeast contribution. However, we do not fully agree with the reviewer that yeast studies are only confirmatory: several aspects, such as the Bax/BclxL association in translocation/retrotranslocation could not have been evidenced in cells where all Bcl-2 family members are present. Also, the ERMES complex is unique to yeast. Alterations of MAM in mammals are ambiguous, because they affect many different processes possibly regulating apoptosis (calcium transport, mitochondrial dynamics, among others). This without mentioning that yeast has been the first model where the capacity of Bax to release cytochrome c was unambiguously demonstrated.

The strain and mutants in Figure 1 are unclear. Please describe in detail.

Also in agreement with reviewers 1 and 2, we have completed the legend of the figure

What is “ERMES” (Line 367) abbreviated?

We have included the abbreviation

Round 2

Reviewer 1 Report

This revised manuscript has been carefully edited to resolve unclear sentences, to include additional interesting information, to improve the figures and many other details, including a very useful new figure 1. There are a few minor details that 'type setters' can deal with (e.g. 3x "et al." instead of "etc." in section 1; a truncated reference number "[105stable" in section 2.2), as well as some incorrect grammar in the Abstract can be corrected as: "allows the investigation of"; last sentence: "members" should be "member" (which otherwise requires the possessive form: "members' localization", or "member localizations", if preferred), and "evidence" ('evidences' is not an official word). 

Author Response

We have corrected all the mistakes noted by the reviewer.

In addition, we have screened the whole manuscript to correct several remaining mispellings and formatting errors.

Reviewer 2 Report

The authors have addressed my previous comments satisfactorily and have improved the manuscript.

Author Response

We have screened the manuscript to correct several additional mispellings and formatting errors.